# Western Diet Modifies Platelet Activation Profiles in Male Mice

**DOI:** 10.3390/ijms25158019

**Published:** 2024-07-23

**Authors:** Adam Corken, Elizabeth C. Wahl, James D. Sikes, Keshari M. Thakali

**Affiliations:** 1Department of Pediatrics, University of Arkansas for Medical Sciences, Little Rock, AR 72205, USA; acorken@uams.edu (A.C.); wahlec@archildrens.org (E.C.W.); sikesjd@archildrens.org (J.D.S.); 2Arkansas Children’s Nutrition Center, Arkansas Children’s Research Institute, Little Rock, AR 72202, USA

**Keywords:** platelets, clotting, cardiovascular disease, western diet, obesity

## Abstract

The correlation between obesity and cardiovascular disease has long been understood, yet scant investigations endeavored to determine the impact of an obesogenic diet on platelet activation or function. As platelets drive clot formation, the terminus of cardiovascular events, we aimed to elucidate the longitudinal effect of an obesogenic diet on platelet phenotype by assessing markers of platelet activation using flow cytometry. Male, weanling mice were fed either a Western diet (30% kcal sucrose, 40% kcal fat, 8.0% sodium) or Control diet (7% kcal sucrose, 10% kcal fat, 0.24% sodium). At 12, 16 and 20 weeks on diets, platelets were collected and stained to visualize glycoprotein Ibα (GPIbα), P-selectin and the conformationally active state of α_IIb_β_3_ (a platelet specific integrin) after collagen stimulation. At all time points, a Western diet reduced GPIbα and α_IIb_β_3_ expression in platelets broadly while P-selectin levels were unaffected. However, P-selectin was diminished by a Western diet in the GPIbα^−^ subpopulation. Thus, a Western diet persistently primed platelets towards a blunted activation response as indicated by reduced active α_IIb_β_3_ and P-selectin surface expression. This study provides a first look at the influence of diet on platelet activation and revealed that platelet activation is susceptible to dietary intervention.

## 1. Introduction

For decades, the link between obesity and cardiovascular disease has been widely known [1,2,3,4]. Moreover, the causal relationship between poor nutrition and the induction and maintenance of an obesogenic state has likewise been well characterized [5,6]. Despite this, there has been a paucity of research into the potential effect that dietary patterns underlying obesity may have on platelet activation and function. As the cellular effector of severe cardiac events such as myocardial infarction and stroke, platelet activation and function is of key interest to investigate [7,8]. Many studies have outlined the mechanisms underlying the pathophysiology of atherogenesis and arterial plaque rupture which in turn instigates platelet activity manifesting in occlusive clot formation [9,10]. However, this causal understanding of the platelet’s terminal role in cardiac events does not address whether an obesogenic diet acts on platelets prior to a vessel rupture to modulate platelet status to potentially modulate the severity of clotting.

To date, a handful of studies have attempted to elucidate circulating platelet status in an obese cohort indirectly by measuring levels of serum markers tangentially associated with platelet activation. While this indicated scholarly movement toward studying the platelets in the context of obesity, none attempted to view the platelet directly. A few additional studies furthered this endeavor by categorizing an increase in the platelet surface receptor glycoprotein VI (GPVI) in both obese humans and overweight mice [11]. Likewise, an increase in GPVI downstream signaling elements was noted in both human and rodent platelets [12]. Though noteworthy, the status of GPVI is not widely considered a barometer of activation. Additionally, the observed signal transduction element participates in several receptor pathways, making it difficult to attribute signaling to GPVI alone [13,14]. And so, we set forth to continue this trajectory by more adequately characterizing the potential impact of an obesogenic diet on the platelet phenotype.

For the present study, we utilized a diet high in fat, sugar and salt in lieu of the customary high fat diet to better recapitulate the dietary intake pattern associated with obesity in humans. Dietary administration was carried out for an extended period of time (20 weeks) with blood sampling occurring at multiple time points in order to capture a longitudinal assessment of the platelet phenotype. The platelet status was assessed by noting the surface expression of three receptors: GPIbα, α_IIb_β_3_ and P-selectin. Platelet activation is marked by a spectrum of events such as the shedding or internalization of GPIbα, conformational shift of the integrin α_IIb_β_3_ and increase in P-selectin via degranulation [15,16,17,18]. Our analysis therefore highlights three different activation characteristics of the platelet in the context of a prolonged diet-driven obesogenic environment.

## 2. Results

### 2.1. Body Mass and Composition Changes Resulting from a Western Diet

Over the course of 20 weeks on respective diets, both groups gained weight, with the Western diet group accruing statistically more weight than the Control diet, becoming evident at 16 weeks on the diets (Figure 1A). Despite this, comparison of lean and fat mass composition via MRI indicated that a Western diet resulted in a dramatic increase in the proportion of fat mass accumulated over the study duration with lean mass being reduced in conjunction (Figure 1B,C). These results indicate that the Western diet is indeed eliciting a heightened obesogenic state relative to the Control diet.

### 2.2. Platelet Receptor Expression Changes in the Total Population as a Result of the Western Diet

Gating onto the platelet population within each sample allowed for the calculation of the percentage of positively fluorescing/expressing platelets as a proportion of the total platelet population for each marker (Figure 2A). Following the establishment of the platelet population and exclusion of all other events, the proportion of positively staining platelets was selected and referenced to the total population in order to determine the overall percentage of positively fluorescing platelets (Figure 2B). The validity of collagen-induced platelet stimulation was determined (Appendix A). Furthermore, we compared vehicle-treated samples to those treated with collagen during the first wave of sample analysis at 12 weeks (Appendix A). This indicated that diet itself was not a driver of platelet activation but that it did influence the platelets’ sensitivity to activation following the introduction of an agonist. It is important to note that P-selectin expression was detected in platelet samples even under basal conditions. Though every methodological precaution was observed, this is an unavoidable phenomenon resulting from the blood collection step necessary to obtain our samples [19]. This phenomenon has been previously observed in multiple models and despite this, dietary and agonistic effects are still capable of elucidation under these conditions as indicated in the Appendix A [20,21,22,23,24,25].

At every time point, a Western diet diminished the percentage of platelets expressing GPIbα on their surface following collagen stimulation (Figure 2C–E). Interestingly, GPIbα surface expression is reduced with platelet activation; this is most often the result of shedding elicited by the protease ADAM17 though other mechanisms such as internalization have been noted as well [26]. Regardless of the specific mechanism, reduced GPIbα is synonymous with platelet stimulation, meaning that a Western diet actually increased activation when characterizing platelets with this marker [27]. Western diet consumption also led to a decrease in the percentage of platelets expressing the active conformation of the integrin α_IIb_β_3_ as denoted by the fluorescent profile of the antibody JON/A [28]. Additionally, a Western diet had no discernable impact on P-selectin expression.

### 2.3. Receptor Expression within Stimulated Platelet Subpopulations Influenced by Western Diet

While we did note sustained changes in the presence of certain platelet markers induced by diet, our previous percentage assessments indicated only the presence and not degree of receptor expression. Thus, we next chose to assess the amount of active α_IIb_β_3_ present on positively expressing platelets to determine if a Western diet also affected this receptor’s abundance in conjunction with reducing its total expression in the platelet population. Gating onto the JON/A^+^ population excluded all platelet events not displaying active α_IIb_β_3_ (Figure 3A), which then allowed for the determination of the receptor’s relative expression within this population (active α_IIb_β_3_^+^) as indicated by the fluorescent mean (Figure 3B). Platelets from mice fed Western diets did indeed demonstrate a reduction in JON/A fluorescence in this population, indicating that the Western diet was additionally diminishing α_IIb_β_3_ expression within the JON/A^+^ positive population. While continuing to focus on the JON/A^+^ population, we next sought to determine if expression varied for other receptors within this subgroup specifically. As such, there was no persistent difference in GPIbα expression within this population (Figure 3C) [as GPIbα negativity is the indication for activation, we could only investigate the percentage of GPIbα-expressing cells and not receptor abundance, as there is no measurable spectrum for negative fluorescence]. Similar to the total platelet population, we saw no difference in the percentage of P-selectin^+^ cells within the α_IIb_β_3_^+^ population and likewise saw no difference in P-selectin surface abundance as indicated by mean fluorescence (Figure 3D,E).

We similarly chose to investigate receptor expression profiles within the GPIbα^−^ population. As mentioned previously, we could not ascertain the degree of GPIbα expression within this subpopulation as we did with α_IIb_β_3_ because the loss of GPIbα denotes the stimulated population and there is no spectrum for negative fluorescence. Despite this, we were able to analyze the profiles for both α_IIb_β_3_ (JON/A) and P-selectin within the GPIbα^−^ population (Figure 4A). We found that similar to the total platelet population, the GPIbα^−^ population demonstrated a reduction in the percentage of α_IIb_β_3_ positive cells as well as reduced α_IIb_β_3_ surface expression in samples from the Western-diet-fed animals (Figure 4B,C). Unlike the total platelet population and the α_IIb_β_3_^+^ pool, the Western diet GPIbα^−^ platelets exhibited a reduction in the percentage of P-selectin^+^ cells, although the relative surface expression of P-selectin was unchanged (Figure 4D,E).

### 2.4. Body Mass Demonstrates No Correlation with GPIbα or α_IIb_β_3_ Expression

While body mass measurements were not different between the Western-diet-fed group relative to the Control group, we examined if there was an overall trend in platelet receptor expression dependent on body mass. Accordingly, we plotted the percentage of positive receptor expression for the total platelet population against matched body mass measurements at 12, 16 and 20 weeks on the respective diets. Since body mass increased for both groups in relation to time, time was therefore a factor entangled with each data point, and sampling periods were combined in a singular assessment as the temporal-body mass association should naturally stratify data points along this axis. Matched data from all three time points were graphed on independent scatterplots in for GPIbα (Figure 5A,B) and JON/A (Figure 5C,D) percent positivity. With regard to the relationship between body mass and active α_IIb_β_3_ expression, there was no demonstrable trend for either the Control or Western diet groups. Similarly, there is no correlation between body mass and GPIba expression for the Control diet platelets. On the other hand, there was a demonstrable trend towards increased GPIbα expression for the Western diet platelets as body mass increases but the small R^2^ value indicates a high degree of variation in the model prediction. Thus, while there is a significant correlation between GPIbα expression and body mass, the statistical modeling is unable to account for a high degree of variation in the prediction of GPIbα levels that is independent of body mass.

### 2.5. Fat Mass Does Not Correlate with Platelet Receptor Expression

The ingestion of an obesogenic diet which in turn causes a heightened accumulation of fat mass presents several avenues wherein our experimental conditions could influence the platelet activation phenotype. Most notably, increased fat accumulation results in heightened systemic inflammation which platelets are keenly influenced by, due to their increasing recognition as an extension of the immune system [29,30,31,32]. And so, if the observed platelet phenotypes are the result of inflammation, which in turn is proportional to fat mass increases, then platelet receptor levels should theoretically correlate to fat composition. Thus, stratification of GPIbα and α_IIb_β_3_ expression should elicit a distribution trend proportional to fat mass composition with the elevated fat mass of Western diet mice being responsible for the observed differences between groups. Similar to our assessment of the relationship between body mass/receptor expression, time was a variable inherently incorporated into each data point as fat mass increased with the progression of time. Therefore, all time points were overlaid on the same scatterplot with the temporal–fat mass association providing a natural indication of the variable of time graphically (Figure 6). No significant correlation was found between fat mass percentage and GPIbα expression for the Control diet platelets (Figure 6A). Similar to our investigation into body mass, there was a significant predictive trend for the Western diet modelling of fat mass in relation to GPIbα expression (Figure 6B). However, we again found the model to be unable to account for a high degree of variation, meaning any predictive interpretations from this apparent trend would be subject to high variability. No significant correlation between JON/A staining and fat mass for either group was found (Figure 6C,D). Overall, this suggests that diet-induced increases in fat mass were not a significant driver of the observed phenotypes.

## 3. Discussion

Our findings indicate that a Western diet facilitates a significant and persistent shift in the platelet activation profile. Indeed, our initial evaluation revealed that the obesogenic diet reduced active α_IIb_β_3_ and GPIbα in the platelet population. While a reduction in receptor expression usually correlates with diminished activation, GPIbα represents an outlier, as this receptor is cleaved, and thereby reduced, as a result of activation. Thus, a Western diet did not universally reduce platelet activation but rather constituted a bifurcating event that favored a GPIbα reduction modality over integrin activation. Moreover, this dietary effect remained constant across multiple sampling periods. Furthermore, we did not limit our pursuit to only investigating these dietary effects at the limited depth of the broad platelet population but instead also investigated platelet subpopulations to more adequately characterize the influence of a Western diet. By focusing specifically on a population positive for one receptor, we could elucidate the emergence of platelet subpopulation by evaluating the expression of the remaining receptors.

The paradigm of platelet activation is nuanced and does not always present in the clear context of “active” or “inactive” [33]. Similarly, platelet activation does not always constitute a sliding scale of clear, sequential, intermediary steps [34]. What has been most investigated is the state of terminal platelet activation such as that found within the nucleus of a growing thrombus (clot) located at the site of vascular injury characterized by platelet degranulation and subsequent P-selectin expression on the membrane surface [35,36]. Prior to this, platelet integrins shift conformation and GPIbα is cleaved from the surface but there is no consensus of association or order of these events nor about their exact proximity to terminal activation. Furthermore, platelet activation has rarely been studied in the context of dietary influence despite the overwhelming understanding of the interplay between diet, obesity and cardiovascular disease. Thus, our findings demonstrate both a relevant characterization of the platelet activation milieu and the impact of an obesogenic state on the platelet activation phenotype.

To date, there has been a limited body of literature attempting to elucidate the effect of poor nutrition on platelet physiology. A few studies attempted to indirectly derive the platelet phenotype by correlating the presence of circulatory markers in obese individuals with the platelet activation status [37,38,39]. Another investigated the transcriptome of platelets from obese individuals, but it is worth noting that the tempo of platelet activation/function has been dictated by circumstances of vessel injuries to be a rapid process that might not be captured by transcriptomic analysis [40]. As such, platelets are primarily reliant on surface receptors already present and granule cargo previously packaged during the platelet’s formation to function adequately, whereas de novo protein synthesis is not a significant factor in platelet functionality [41]. However, relatively recently, there has been a series of investigations by a singular group to directly assess platelet phenotype associated with obesity. These attempts were somewhat narrow, as their focus was limited to the receptor GPVI, which was found in higher abundance on platelets from individuals with obesity [12]. Likewise, the investigators also found that obesity increased platelet Src phosphorylation, which is a known downstream signal transducer of GPVI [42]. This study was followed by a rat model wherein high-fat-diet-fed animals presented marginally higher platelet binding to collagen, the GPVI ligand, as well as similarly higher levels of Src phosphorylation [11]. However, Src is a redundant component of signal transduction for several platelet receptors, and collagen is likewise the ligand of the adhesion receptor a_2_b_1_ which platelets also express [43,44]. Furthermore, both human and animal samples were analyzed at only a singular time point, which failed to provide a temporal framework for identifying the modulation of the platelet phenotype. Thus, while providing a necessary starting position for characterizing the relationship between diet and platelets, there is much to be explored and refined to provide a better foundation of understanding. As such, we have documented the expression of multiple receptors with expression profiles that have clear precedent for relating to platelet activation. Additionally, our analysis was conducted at three time points separated by several weeks to provide a longitudinal assessment of receptor fluctuations. Finally, our model utilized a diet with elevated fat, sugar and salt levels instead of a diet singularly high in fat to more adequately recapitulate the dietary intake of humans which contributes to an obesogenic state.

Accordingly, we noted that a Western diet not only diminished the overall percentage of GPIbα^+^ platelets but also reduced active α_IIb_β_3_ expression within the GPIbα^−^ population (Figure 7). Similarly, P-selectin expression within the GPIbα^−^ subpopulation was reduced with a Western diet (whereas P-selectin expression exhibited no change as a result of diet when viewing the broad platelet population). This means that although a Western diet facilitates a greater proportion of activation through the GPIbα reduction mechanism (as a Western diet exhibits a greater percentage of GPIbα^−^ platelets), these platelets may overall be less stimulated than the Control diet counterparts which contain higher thresholds of α_IIb_β_3_ activation and P-selectin expression. We did not inversely see a difference in GPIbα expression when prioritizing the [active] α_IIb_β_3_ population for comparison between the two groups. The possible explanation for these phenomena is that regardless of diet, the GPIbα^−^ population constitutes a small percentage while the [active] α_IIb_β_3_ population is much larger relative to the total platelet population. And so, since the α_IIb_β_3_^+^ population differed as a result of diet in the broad platelet population, the small GPIbα^−^ had a decent probability of overlapping with the α_IIb_β_3_^+^ population and exhibiting a comparable receptor expression profile to the entire platelet population for α_IIb_β_3_. Inversely, the overwhelming majority of platelets are GPIbα^+^, meaning there is a large chance the α_IIb_b_3_^+^ population will overlap with GPIbα^+^ and not GPIbα^−^ platelets and this may explain why α_IIb_β_3_^+^ platelets do not differ in GPIbα^−^ content across diets. While this constitutes a reasonable explanation for the relationship of GPIbα/α_IIb_β_3_, it does not adequately account for the observed reduction in P-selectin expression within the GPIbα^−^ subpopulation, indicating that diet exerts a multifactorial manipulation of receptor expression within the context of platelet activation that results in the appearance of several subcategories of platelet phenotype maintained over time.

While this study presents a necessary foundational step towards better understanding the relationship between diet and platelet activation, the findings are not without limitations. First, the data represent the platelet phenotype following collagen stimulation; therefore, it is presently unknown if diet affects the baseline platelet phenotype additionally. Additionally, due to time and budgetary constraints, the studies were only performed in male mice; thus, it is unknown if our findings hold true for both sexes. Moreover, while we have attempted to address the mechanisms behind the shift in platelet phenotype, additional assessments could be more refined. Future studies are needed to determine whether dietary metabolites and/or induced inflammation or other mechanisms are directly or indirectly responsible for the platelet changes we observed.

## 4. Materials and Methods

### 4.1. Mouse Model and Diet Administration

Four-week-old male C57Bl/6J mice were obtained from Jackson Laboratory (Strain #000664). Upon receiving, animals received microchip identifiers and were subsequently randomized into groups and fed ad libitum either a Control (Research Diets Inc. (New Brunswick, NJ, USA) D12450J; 7% kcal sucrose, 10% kcal fat, 0.24% sodium) or Western diet (Research Diets Inc. D06111701; 30% kcal sucrose, 40% kcal fat, 8.0% sodium) for 20 weeks, with 20 total animals per diet group. Additionally, mice were weighed and body composition was assessed using an EchoMRI (EchoMRI LLC. (Houston, TX, USA) Model# EMR-035) upon receiving. Body mass was recorded weekly throughout the study duration while MRI was performed at 12, 16 and 20 weeks of dietary administration. The Institutional Animal Care and Use Committee at the University of Arkansas for Medical Sciences approved all experimental procedures.

### 4.2. Blood Collection

Whole blood was collected from the retro-orbital sinus at 12, 16 and 20 weeks following diet administration. Blood droplets were collected from the capillary tube into 1.5 mL centrifuge tubes containing 3.8% sodium citrate at a ratio of 1:10 the total volume of blood. Approximately 100 μL of blood was collected from each mouse at each time point.

### 4.3. Platelet Rich Plasma Generation and Stimulation

Platelet rich plasma (PRP) was generated from whole blood by centrifuging at 200× *g* for 20 min at room temperature. PRP constituted the top layer of the fractionated blood. 25 μL of PRP was then diluted with an equal volume of PBS in a 5 mL round bottom tube and placed in a 37 °C water bath and allowed to acclimatize for 10 min. Collagen (Chrono-log P/N 387) was then added to a final concentration of 20 μg/mL and stimulation proceeded for 15 min. Afterward, the tube was removed from the water bath and antibody staining conducted.

### 4.4. Flow Cytometry Preparation, Acquisition and Analysis

Stimulated PRP was stained with 50 μL of a primary labeled antibody cocktail containing anti-P-selectin (BD Biosciences (Franklin Lakes, NJ, USA) 740884, BV785), anti-GPIbα (Emfret (Wurzburg, Germany) M040-1, FITC) and anti-α_IIb_β_3_ [active] (Emfret (Wurzburg, Germany) M023-2; antibody name—JON/A, PE) diluted in PBS. Staining proceeded for 20 min with constant gentle agitation followed by the administration of a fixation solution (BD FACS Lysing Solution #349202). The samples were again agitated gently for 10 min to allow for the fixation of samples and lysing of any red cell contaminants. Samples were then refrigerated (4 °C) until data acquisition. A BD LSRFortessa was used for data collection with the acquisition gate set to collect 100,000 platelet events per sample. The fluorescent signal for platelet-specific events was recorded as both the geometric mean and percentage of the designated population. FlowJo^TM^ software (v10.9.0) was used for the analysis of acquired flow cytometry data. Data were plotted using GraphPad Prism^TM^ software (v10.2.3) to express mean values ± SEM for each graph.

### 4.5. Statistical Analysis

GraphPad^TM^ Prism was used for statistical analysis of all data. Specifically, a 2-way ANOVA was used for the comparison of body mass and fat/lean mass accumulation. A Mann–Whitney U test was used in the comparison of receptor expression values. Pearson’s correlation was used to describe the association between body mass, fat mass and platelet receptor expression data points.

## 5. Conclusions

The results of our study indicate that an obesogenic state elicited by Western diet feeding in male mice causes changes in the platelet activation profile. Western diet consumption generated a significant reduction in cells with active surface integrins (α_IIb_β_3_) and the adhesion receptor GPIbα. This implies that a Western diet causes a divergence in platelet activation favoring the pathway(s) leading to GPIbα reduction over those underlying integrin activation. Interestingly, although a Western diet generates more GPIbα^−^ platelets, further analysis of this subcategory revealed a reduction in P-selectin and a_IIb_β_3_ expression, suggesting that these cells are less active than in the Control-fed counterparts. These findings provide a necessary foundation for further investigations into how diet composition modifies the molecular milieu regulating platelet phenotype and the pathophysiological consequences of a high fat diet on platelet activation and function.

## Figures and Tables

**Figure 1 ijms-25-08019-f001:**
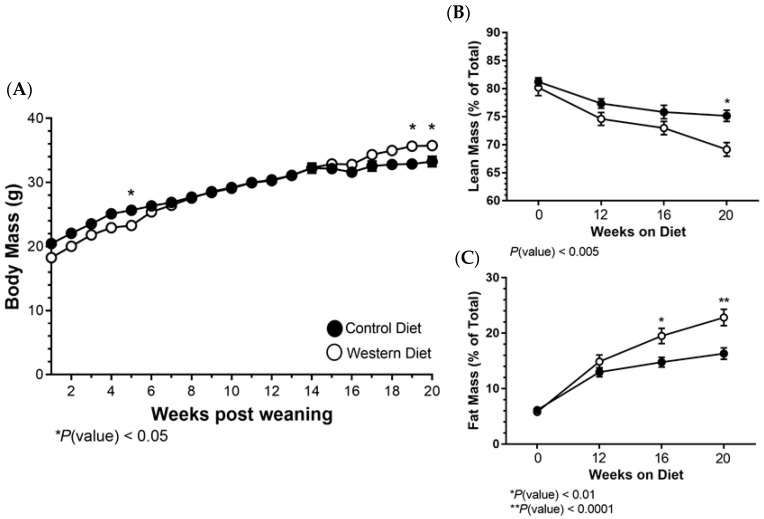
(**A**) Body mass was measured beginning at the time of dietary administration until the study conclusion. (**B**,**C**) Lean and fat mass was assessed prior to diet administration and at 12, 16 and 20 weeks via MRI. The data are represented as the mean ± SEM. * or ** represents a statistically significant difference between groups as noted on each graph.

**Figure 2 ijms-25-08019-f002:**
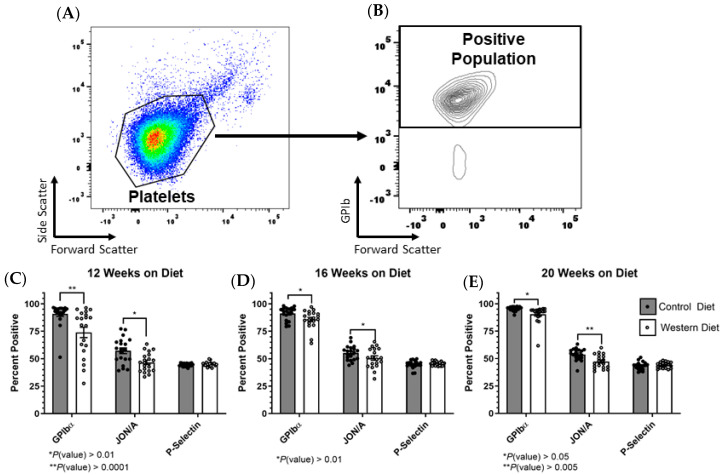
(**A**) Using forward and side scattering axes, the platelet population was gated upon. (**B**) Within the platelet population, the positive staining population was determined for each fluorescent marker. (**C**–**E**) The percentage of the total platelet population positively fluorescing for GPIbα, α_IIb_β_3_ [JON/A] and P-selectin was calculated across all three time points. The data are represented as the mean ± SEM.

**Figure 3 ijms-25-08019-f003:**
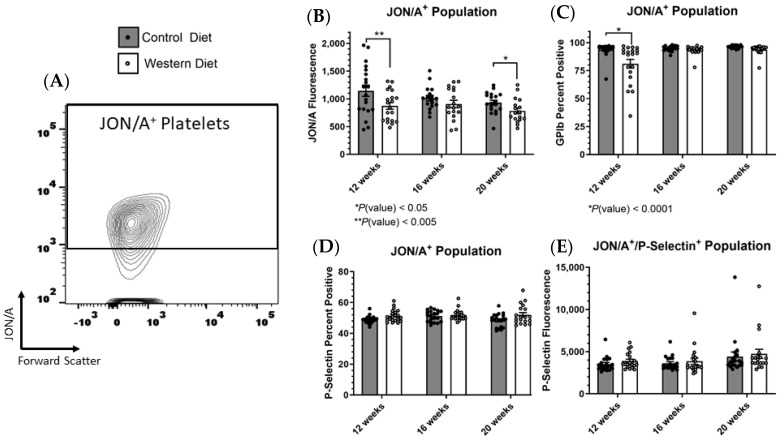
(**A**) The JON/A^+^ (active α_IIb_β_3_^+^) population was determined and selected for further analysis. (**B**) Within this subpopulation, the JON/A mean fluorescence was calculated for all three time points. (**C**) The percentage of GPIbα and (**D**) P-selectin-positive cells as well as (**E**) average P-selectin fluorescence were also analyzed within the JON/A^+^/P-selectin^+^ subpopulation. The data are represented as the mean ± SEM.

**Figure 4 ijms-25-08019-f004:**
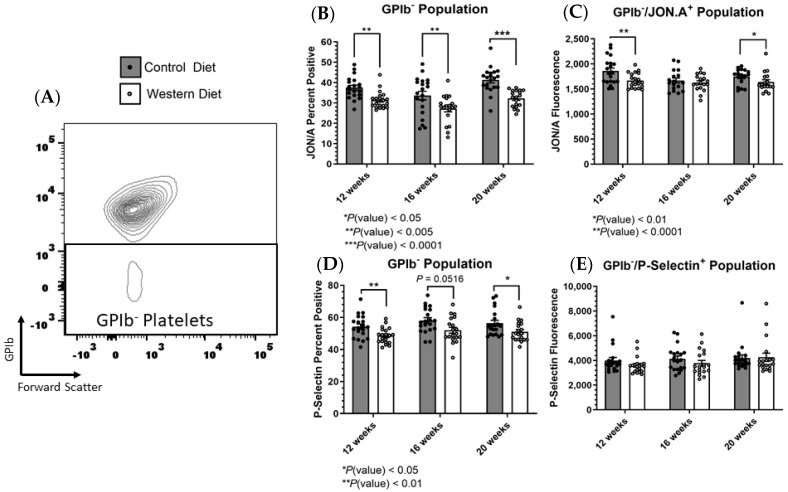
(**A**) The GPIbα^−^ was established to exclude all other events. (**B**) The percentage of [active] α_Iib_β_3_^+^ cells and (**C**) the mean fluorescence of α_Iib_β_3_ were recorded for the GPIbα^−^ platelet pool for the three established time points. (**D**) The proportion of P-selectin^+^ platelets in conjunction with (**E**) the mean fluorescence of P-selectin was also determined within the GPIbα^−^ subpopulation. The data are represented as the mean ± SEM.

**Figure 5 ijms-25-08019-f005:**
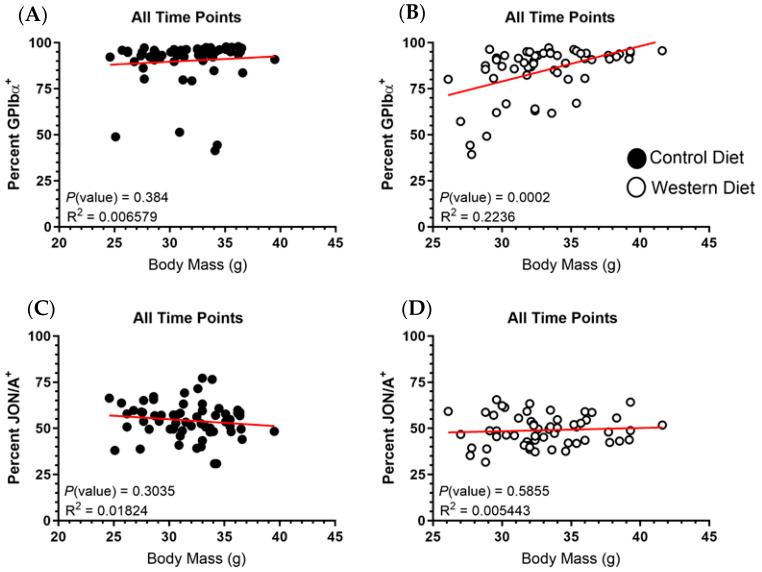
The percentage of GPIbα for the Control (**A**,**B**) and Western diet groups and [active] α_IIb_β_3_ (**C**,**D**) positive platelets for each diet cohort was plotted in relation to body mass measurements. The data points from all three time points (12, 16 and 20 weeks on diet) were graphed together as one dataset. A line of best fit colored red is present within the dataset on each graph.

**Figure 6 ijms-25-08019-f006:**
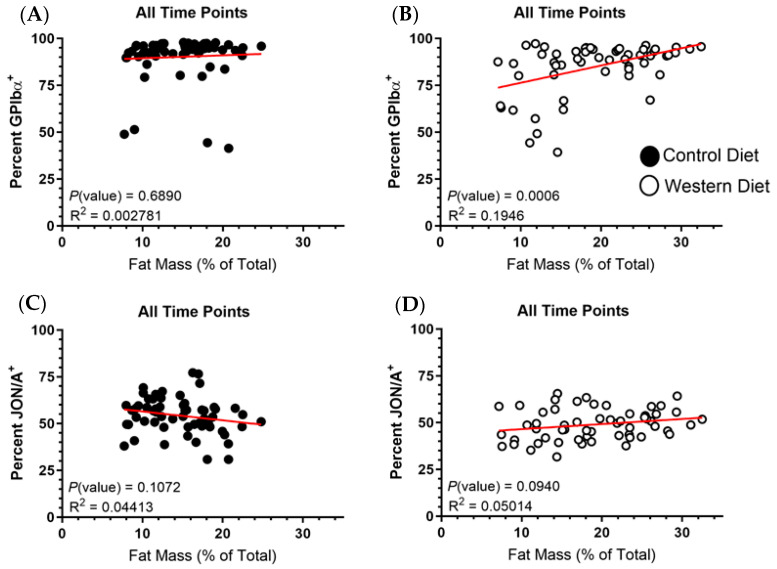
The percentages of GPIbα for Control (**A**) and Western (**B**) positive platelets as well as for [active] α_IIb_β_3_ (**C**,**D**) were graphed in relation to fat mass measurements. All data points were overlaid on the same graph, regardless of the time period in which they were collected. A line of best fit colored red is present within the dataset on each graph.

**Figure 7 ijms-25-08019-f007:**
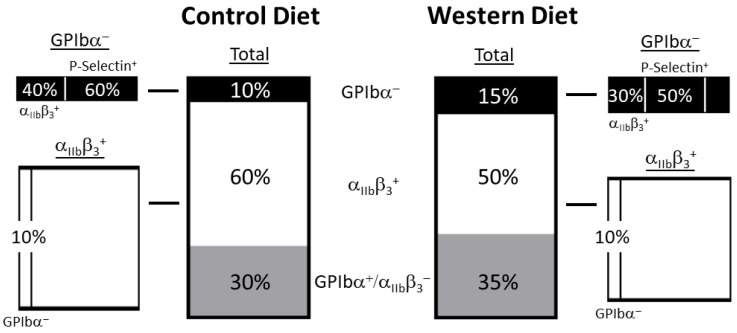
The composition of receptor expression profiles for Control- and Western-diet-fed animals following collagen stimulation reveals differences in GPIbα and [active] α_IIb_β_3_ positive cell composition in the total platelet population. Analysis within the GPIbα^−^ and [active] α_IIb_β subpopulations revealed compositional differences in α_IIb_β_3_ and P-selectin expression in the GPIbα^−^ population only.

## Data Availability

The dataset outlined in this manuscript is available from the corresponding author upon request.

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
