# Peer review of "Western Diet Modifies Platelet Activation Profiles in Male Mice"

_ijms, 2024, doi:10.3390/ijms25158019_

Round 1
Reviewer 1 Report
Comments and Suggestions for Authors
In the current manuscript, Corken et al. monitor the impact of high-fat/sucrose/salt diet (Western diet) on three selected surface receptors (P-Selectin, active GPIIb/IIIa and GPIbalpha) on collagen-activated murine platelets. While P-selectin levels were unaltered by Western diet, active GPIIb/IIIa and GPIbalpha were diminished, the latter of which the authors attribute to GPIbalpha shedding. By combining platelet positivity for the different receptors, platelet subpopulations were identified. The authors conclude that Western diet induces ambiguous changes in platelet function, with indications for both hypo- (active GPIIb/IIIa) and hyper-responsiveness (GPIbalpha).
General comments:
The study is interesting, novel and provides essential investigations into the direct effects of diet on platelet function, a topic that was previously neglected when examining obesity-associated thrombotic/cardiovascular risk.
While the aim and content are highly relevant, the study has some weaknesses, the main being that 1) no data on basal receptor levels are provided and 2) increase in GPIb-negative events is overinterpreted as platelet receptor shedding and platelet activation, though (partial) loss of GPIb could have numerous causes, such as receptor internalisation, extracellular vesicle release, defective GPIb packaging by megakaryocytes, … (DOI: 10.1186/s12959-019-0209-5).
1) What is the effect of Western diet on basal CD62P, GPIIb/IIIa and GPIb? Low GPIIb/IIIa levels following stimulation do not categorically indicate reduced function, but could also be a consequence of pre-activation and platelet exhaustion. Evaluation of basal levels is thus crucial to accurately interpret the presented observations.
2) GPIb shedding must be confirmed, e.g. by measuring glycocalicin in plasma or supernatant of stimulated platelets.
Specific comments:
1) lines 88: Were primary antibodies directly labelled? If so, fluorophores should be provided. If not, information on secondary antibodies should be provided.
2) lines 99-104: The authors should explain what the presented values and error bars denote. SD should be preferred over SEM. Given the use of Mann-Whitney U test, which is appropriate for non-normally distributed data, data presentation as mean and SD (or SEM) is not ideal, but rather median and range (or quantile).
3) Fig. 1: WD mice had lower start weight than Control mice, resulting in little weight difference after 20 weeks despite clearly different body composition. Additional evaluation of % weight change may better represent diet-induced changes and also help with correlation analyses in Fig. 5. Does HFD change platelet count or mean platelet volume, which would hint at changes in thrombopoiesis?
4) Lines 124-126: As stated in the general comments above, “Western diet actually increased activation” is an overinterpretation. The authors need to confirm this conclusion by showing basal activation levels, plasma and/or stimulation-supernatant glycocalicin.
5) Lines 150-153 (and 163-164): Evaluation of GPIbalpha MFI on GPIb-negative cells obviously makes no sense. However, GPIb shedding (or internalisation etc.) on a whole platelet level is not a yes/no event, but a graded process. As such, analysis of MFI within the GPIb-positive population could provide insight into the relative changes.
6) Fig. 3A: The gate identifying JON/A+ platelets transects the platelet population despite the presence of a separate population at the bottom of the plot. The authors should also provide a representative plot for non-activated and/or unstained platelets to confirm the accuracy of gating.
7) Line 159: As indicated in the graph, E) analyses P-selectin MFI in P-sel+/JON/A+ double positive cells, not just JON/A+ cells. Please correct the text accordingly.
8) Fig. 4A: Identification of platelets merely by FSC and SSC entails the risks of falsely including e.g. cellular debris, particularly after potent stimulation. How can the authors be sure that GPIb-negative events are platelets? Are they positive for CD41 or CD61? How does the plot look without collagen stimulation?
9) Line 176: With an R2=0.055, the correlation of %GPIbalpha+ and Body mass is really quite weak and not “moderate”.
10) Fig. 5+6: Can the authors provide separate curves for control and Western diet? As mentioned above, GPIbalpha MFI might allow a more nuanced evaluation of receptor expression than the very black-and-white %.
11) Line 280: Do the authors mean “diminished the overall percentage of GPIbalpha-POSITIVE platelets”?
12) Fig.7: This graph is not very clear, starting at the order of legend labels but also arrows going through elements or texts. As it represents a summary of observations, the legend should clearly state that these values represent activated platelets, not basal conditions.

Reviewer 2 Report
Comments and Suggestions for Authors
1. The results of this study indicate that a Western diet reduces the expression of platelet GPIbα and αIIbβ3, thereby inhibiting platelet activation. This leads to a decrease in platelet adhesion ability, inhibiting thrombus formation. However, this contradicts the findings in the previous research reporting that obesity promotes platelet activation. Could this discrepancy be related to the composition or duration of the diet?
2.The evidence for platelet activation in this article is not sufficient. Further experiments such as platelet aggregation tests, clot retraction experiments, and ELISA measurements of plasma fibrinogen or fibrinogen levels could be added to strengthen the evidence.
3. In this paper, the analysis of platelet receptor expression profiles under different dietary conditions is presented. Further analysis could involve studying the platelet receptor expression profiles under different feeding durations.
Author Response
Reviewer # 2
Comment 1: The results of this study indicate that a Western diet reduces the expression of platelet GPIbα and αIIbβ3, thereby inhibiting platelet activation. This leads to a decrease in platelet adhesion ability, inhibiting thrombus formation. However, this contradicts the findings in the previous research reporting that obesity promotes platelet activation. Could this discrepancy be related to the composition or duration of the diet?
Response 1: This is an excellent postulate and one we’ve contemplated much ourselves as we interpreted the data. Undoubtedly the dietary composition could be a factor in eliciting these results. While it was designed with the intention of recapitulating the processed diets that go hand-in-hand with obesity and the heightened cardiovascular disease risk, it undoubtedly lacks all the intricacies of real-world dieting patterns. It is possible that the utilization of a murine model more responsible for these results appearing in contradiction to previous reports as it is not uncommon for mice and human biology failing to properly synchronize under laboratory conditions. Additionally the strain of mouse utilized is undoubtedly a significant factor. Thus, the diet and the model organism are the most plausible explanations for this discrepancy if the results of our experiments are perceived as artifacts. Alternatively, there are potential explanations in support of the opposing notion that the results are true phenomenon.
First, it is important to note that a reduction in GPIba is a sign of platelet activation so in this manner Western diet platelets demonstrate an increased activation marker relative to the Control diet group. Furthermore, while Control diet samples have increased active aIIbb3, Western diet samples still do express the active form of this receptor (albeit at lower levels). When taken together this shows that Western diet platelets shows heightened activation in one instance (reduced GPIba) and reduced in another (reduced aIIbb3); and so, Western diet platelets are still overall active. In the absence of data from future planned experiments, we hypothesize that Western diet platelets will still activate/aggregate and form a thrombus in the presence of a stimulus but the rate of formation, thrombus size, etc. may be reduced relative to Control diet platelets. In future experiments we plan to test this hypothesis with in vivo experiments highlighted in our response to the second comment. While this appears to be contradictory to current knowledge, we believe a nuanced view of the present literature supports our findings.
Second, current literature investigating obesity and platelets directly appears split with reports both indicating platelet reactivity is unaffected (10-12) or heightened activation. Of those that show increased activation, several assay using aggregometry that offers relatively weak results (13-15). Additionally, as outlined in our response to the second comment, aggregometry can be a problematic assay. Other studies supporting increased activation, soluble markers in the serum were assayed which does not provide insight into the platelet phenotype directly as we have with flow cytometry (16). Of note, one study in support of diet induced increases in platelet activation utilized Ldlr and ApoE knockout animals to increases the severity of dietary influence. This harkens back to our previous sentiment that the model could be a significant contributor to our observed phenotype. And so, we will conduct future experiments with these models to further characterize the influence of diet on platelet phenotype. Nonetheless, the literature focusing on dietary effects within platelets directly does not overwhelmingly refute our findings.
Finally, the relationship between cardiovascular disease (CVD) and obesity is beyond question. The initiation/propagation of atherosclerosis and risk of plaque rupture is clearly linked to bodyweight increases. It is important to note that though platelets are a significant contributor to CVD outcomes, knowledge of the severity of plaque lesions, etc. does not indicate the status of the platelet. So while elevated CVD risk establishes the appropriate environmental conditions for platelet activation, it is does not simultaneously represent a heightened platelet activation phenotype. We hypothesize that both our Control and Western diet platelets will form an occlusive thrombus in the presence of a ruptured vessel due to the litany of agonist present. However, since plaque formation/rupture is predominantly associated with obesity, we only observe the pathophysiological severity of occlusive thrombi formed with obese (Western) platelets under obese conditions. Healthy/control platelets aren’t subject to similar incidences of vessel rupture and so there has not been the appropriate scenario to contrast the activation/functionality of control to obese (Western) under pathophysiological conditions. We are unable to discern if obese (Western) platelets are slightly less reactive relative to healthy controls, only that these platelets can for thrombi under conditions of vessel injury. So knowing at present knowing the association between CVD and obesity does not indicate the status of the platelet as it may be influenced by diet. Thus, we don’t at present know that obese (Western) don’t demonstrate slightly less activation-associated phenotypic characteristics.
In sum, it is more than reasonable to assume that our findings are the result of the diet and mouse model utilized for the study. However, there is a compelling argument that present literature does not refute the eligibility of our findings as accurate.
Comment 2: The evidence for platelet activation in this article is not sufficient. Further experiments such as platelet aggregation tests, clot retraction experiments, and ELISA measurements of plasma fibrinogen or fibrinogen levels could be added to strengthen the evidence.
Response 2: The reviewer provides an excellent point and we will further address platelet functionality in vivo in our next set of follow up experiments using carotid artery occlusion and tail bleeding assays in lieu of in vitro clot retraction experimentation. Regarding our rationale for choosing to characterize platelet activation via flow cytometry, the following logic was utilized:
The most common means for determining platelet activation are either through the 1) change in levels of surface markers or 2) indication of degranulation by the detection of soluble mediators known to be stored in platelet granules. Flow cytometry readily allows for the first method of determining platelet activation as it permits the quantification of multiple surface markers in a single sample. As such, the utilization of flow cytometry for the determination of the platelet’s activation status has become increasingly common as it allows for the direct assessment of the platelet population (17-20).
Prior to the advancement of flow cytometry and the commercial availability of platelet specific antibodies for labeling, aggregometry was the primary means for determining platelet activation indirectly. Comparisons between flow cytometry and aggregometry assays indicate that flow cytometry equal in detecting platelet activation status relative to aggregometry (21). However, due to the reduced sample processing workflow and reduced sample volume requirement, flow cytometry proved superior for practical purposes. Additionally, flow cytometry is capable of more sophisticated analysis such as the determination of the specific activation status of certain integrins (aIIbb3) through the commercial availability of highly specific antibody clones (ex. JON/A – Emfret Analytics) which aggregometry would be unable to address. Moreover, there have been a number of reports indicating that the process of aggregation (platelet adhesion) can occur independent of activation thereby reducing the ability to accurately interpret aggregometry data (22, 23). Furthermore, the process of aggregometry more prone to analytical fluctuations due to a number of known (and unexplained) experimental and environmental conditions (24, 25).
The measurement of fibrinogen via ELISA would strengthen our findings as the reviewer indicated but we are unable to conduct this as the whole blood and plasma fraction extracted for this experiment was completely consumed for other assays. This would indicate platelet a-granule release as fibrinogen is part of the granular cargo but likewise, P-selectin is stored within membrane of these same granule and so surface expression of P-selectin would similarly increase as the granule membrane fused with the plasma membrane during degranulation. Since we measured platelet surface P-selectin during our flow analysis, we theoretically conducted a comparable assay to a fibrinogen ELISA but in a direct rather than indirect manner. As fibrinogen is synthesized in the liver and secreted into the plasma for platelets uptake, a fibrinogen ELISA using plasma could be influenced by residual plasma fibrinogen levels and confound any interpretation linking measurements to platelet degranulation.
Comment 3: In this paper, the analysis of platelet receptor expression profiles under different dietary conditions is presented. Further analysis could involve studying the platelet receptor expression profiles under different feeding durations.
Response 3: This is an excellent comment. The duration of feeding for our study was dependent on a number of experimental and financial considerations, thus it was out of the scope for this preliminary study to several different feeding durations. Future experiments are planned to explore whether our observed discrepancies in platelet characteristics between the two diets becomes more or less pronounced with an extended duration of feeding. As the cardiovascular effects brought about by poor nutrition are the result of prolonged dietary habits, it is imperative that we extend our feeding window in future studies to further elucidate the effect diet may be playing on platelets directly.
References
- Han Y, Nurden A, Combrié R, Pasquet JM. Redistribution of glycoprotein Ib within platelets in response to protease-activated receptors 1 and 4: roles of cytoskeleton and calcium. J Thromb Haemost. 2003;1(10):2206-15.
- Moroi M, Farndale RW, Jung SM. Activation-induced changes in platelet surface receptor expression and the contribution of the large-platelet subpopulation to activation. Res Pract Thromb Haemost. 2020;4(2):285-97.
- Lamrani L, Lacout C, Ollivier V, Denis CV, Gardiner E, Ho Tin Noe B, et al. Hemostatic disorders in a JAK2V617F-driven mouse model of myeloproliferative neoplasm. Blood. 2014;124(7):1136-45.
- Lu Y, Li Q, Liu Y-Y, Sun K, Fan J-Y, Wang C-S, et al. Inhibitory effect of caffeic acid on ADP-induced thrombus formation and platelet activation involves mitogen-activated protein kinases. Scientific Reports. 2015;5(1):13824.
- Corken AL, Ong V, Kore R, Ghanta SN, Karaduta O, Pathak R, et al. Platelets, Inflammation, and Purinergic Receptors in Chronic Kidney Disease. Kidney Int. 2024.
- Bagamery K, Kvell K, Landau R, Graham J. Flow cytometric analysis of CD41-labeled platelets isolated by the rapid, one-step OptiPrep method from human blood. Cytometry A. 2005;65(1):84-7.
- Asfari A, Dent JA, Corken A, Herington D, Kaliki V, Sra N, et al. Platelet Glycoprotein VI Haplotypes and the Presentation of Paediatric Sepsis. Thromb Haemost. 2019;119(3):431-8.
- Croisé B, Paré A, Joly A, Louisy A, Laure B, Goga D. Optimized centrifugation preparation of the platelet rich plasma: Literature review. J Stomatol Oral Maxillofac Surg. 2020;121(2):150-4.
- Dhurat R, Sukesh M. Principles and Methods of Preparation of Platelet-Rich Plasma: A Review and Author's Perspective. J Cutan Aesthet Surg. 2014;7(4):189-97.
- Bukhari IA, Habib SS, Alnahedh A, Almutairi F, Alkahtani L, Alareek LA, et al. Relationship of Body Adiposity with Platelet Function in Obese and Non-obese Individuals. Cureus. 2020;12(1):e6815.
- Samocha-Bonet D, Justo D, Rogowski O, Saar N, Abu-Abeid S, Shenkerman G, et al. Platelet counts and platelet activation markers in obese subjects. Mediators Inflamm. 2008;2008:834153.
- Ezzaty Mirhashemi M, Shah RV, Kitchen RR, Rong J, Spahillari A, Pico AR, et al. The Dynamic Platelet Transcriptome in Obesity and Weight Loss. Arterioscler Thromb Vasc Biol. 2021;41(2):854-64.
- Puccini M, Rauch C, Jakobs K, Friebel J, Hassanein A, Landmesser U, et al. Being Overweight or Obese Is Associated with an Increased Platelet Reactivity Despite Dual Antiplatelet Therapy with Aspirin and Clopidogrel. Cardiovascular Drugs and Therapy. 2023;37(4):833-7.
- Barrachina MN, Sueiro AM, Izquierdo I, Hermida-Nogueira L, Guitián E, Casanueva FF, et al. GPVI surface expression and signalling pathway activation are increased in platelets from obese patients: Elucidating potential anti-atherothrombotic targets in obesity. Atherosclerosis. 2019;281:62-70.
- Barrachina MN, Morán LA, Izquierdo I, Casanueva FF, Pardo M, García Á. Analysis of platelets from a diet-induced obesity rat model: elucidating platelet dysfunction in obesity. Sci Rep. 2020;10(1):13104.
- Davì G, Guagnano MT, Ciabattoni G, Basili S, Falco A, Marinopiccoli M, et al. Platelet Activation in Obese WomenRole of Inflammation and Oxidant Stress. JAMA. 2002;288(16):2008-14.
- R S, Saharia GK, Patra S, Bandyopadhyay D, Patro BK. Flow cytometry based platelet activation markers and state of inflammation among subjects with type 2 diabetes with and without depression. Scientific Reports. 2022;12(1):10039.
- Södergren AL, Ramström S. Platelet subpopulations remain despite strong dual agonist stimulation and can be characterised using a novel six-colour flow cytometry protocol. Scientific Reports. 2018;8(1):1441.
- van Velzen JF, Laros-van Gorkom BAP, Pop GAM, van Heerde WL. Multicolor flow cytometry for evaluation of platelet surface antigens and activation markers. Thrombosis Research. 2012;130(1):92-8.
- Morel A, Rywaniak J, Bijak M, Miller E, Niwald M, Saluk J. Flow cytometric analysis reveals the high levels of platelet activation parameters in circulation of multiple sclerosis patients. Molecular and Cellular Biochemistry. 2017;430(1):69-80.
- Navred K, Martin M, Ekdahl L, Zetterberg E, Andersson NG, Strandberg K, et al. A simplified flow cytometric method for detection of inherited platelet disorders-A comparison to the gold standard light transmission aggregometry. PLoS One. 2019;14(1):e0211130.
- Ruggeri ZM, Orje JN, Habermann R, Federici AB, Reininger AJ. Activation-independent platelet adhesion and aggregation under elevated shear stress. Blood. 2006;108(6):1903-10.
- Jackson SP. The growing complexity of platelet aggregation. Blood. 2007;109(12):5087-95.
- Alessi MC, Sié P, Payrastre B. Strengths and Weaknesses of Light Transmission Aggregometry in Diagnosing Hereditary Platelet Function Disorders. J Clin Med. 2020;9(3).
- Hayward CPM, Moffat KA, Brunet J, Carlino SA, Plumhoff E, Meijer P, et al. Update on diagnostic testing for platelet function disorders: What is practical and useful? Int J Lab Hematol. 2019;41 Suppl 1:26-32.
Round 2
Reviewer 1 Report
Comments and Suggestions for Authors
The author addressed the majority of comments, including important new supplement data on basal and collagen-induced platelet activation in control and 12-week Western diet mice. While the manuscript has greatly improved, there are still some minor aspects that should be addressed:
1) In response to general comment 1, the authors provide additional data in the supplement, depicting the comparison of vehicle vs collagen stimulation on platelet surface markers in naïve control mice (Fig. S1) and following 12 weeks on diet (Fig. S2). While I do not doubt the overall results, the data show very high basal platelet activation with about 40-45% P-selectin and Jon/A-positive platelets already in unstimulated blood of control animals, which suggests problems with platelet preparation or gating, thought the FMO-gating explained by the authors is indeed a good way of gating.
The provided data demonstrates that diet did not affect basal activation, but only responsiveness. This is an important finding as it may have implications for the potential underlying mechanisms. However, in the manuscript text, the authors merely state that they verified collagen stimulation. The text should be adapted to also showcase these important findings on basal activation status.
2) The use of SEM is not appropriate to depict variability of data. I appreciate that the author now depict individual values to overcome this problem. Data presentation in the supplement figures should be adjusted accordingly to also show individual points, not just mean +/- SEM.
3) The authors added a note on statistics and data presentation to Fig. 5 and Fig. 6 that is not reflective of the actual figure, which does NOT show mean and SEM, nor any asterisk. This needs to be corrected.
4) In response to specific comment 3, the authors provide a plot in the CV depicting % weight change. However, the calculations of % weight change appear to be incorrect, as 100% increase in the control group would mean an increase from ~20g starting weight to 40g after 20weeks, which is not in line with the absolute numbers provided in Fig. 1. Values should be re-calculated. Please provide the revised plot in the supplement.
5) The authors explain in the CV why GPIbalpha shedding is the most likely interpretation of reduced surface levels. Lacking evidence of actual shedding due to unavailability of quality assays, but referencing the effect as such without further explanation still remains problematic. The authors should address the different possibilities of GPIbalpha loss in the discussion and provide information in the manuscript as well, why they interpret low GPIbalpha levels as shedding.

Reviewer 2 Report
Comments and Suggestions for Authors
I have no other comments.
Author Response
Thank you for your thorough review.